# A Review of Vehicle-to-Vulnerable Road User Collisions on Limited-Access Highways to Support the Development of Automated Vehicle Safety Assessments

**Husam Muslim** [1,2,*,†] and **Jacobo Antona-Makoshi** [1,†]

1    Japan Automobile Research Institute (JARI), 2530 Karima, Tsukuba 305-0822, Ibaraki, Japan; ajacobo@jari.or.jp
2    Faculty of Engineering, Information and Systems, University of Tsukuba, 1-1-1 Tennoudai,
     Tsukuba 305-0822, Ibaraki, Japan
*    Correspondence: ahusam@jari.or.jp or hussam@css.risk.tsukuba.ac.jp; Tel.: +81-(0)29-856-0838
†    These authors contributed equally to this work.

**Abstract:** This study aims to provide evidence to support the development of automated vehicle (AV) safety assessments that consider the possible presence of non-motorized vulnerable road-users (VRUs) on limited-access highways. Although limited-access highways are designed to accommodate high-speed motor vehicles, collisions involving VRUs on such roadways are frequently reported. A narrative review is conducted, covering the epidemiology of VRUs crashes on limited-access highways to identify typical crash patterns considering collisions severity and the underlying reasons for the VRUs to use the highway. The review results show that occupants alighting from a disabled or crashed vehicle, people seeking help or helping others, highway maintenance zones, police stops, and people crossing a highway should be given priority to ensure VRU safety on limited-access highways. The results are summarized in figures with schematic models to generate test scenarios for AV safety assessment. Additionally, the results are discussed using two examples of traffic situations relevant to the potential AV-VRU crashes on highways and the current performance of autonomous emergency braking and autonomous emergency steering systems. These findings have important implications for producing scenarios in which AV may not produce crashes lest it performs worse than human drivers in the proposed scenarios.

**Keywords:** automated driving systems; collision avoidance; highway; traffic safety; pedestrians

## 1. Introduction

Road traffic crashes involving non-motorized VRUs, including pedestrians and cyclists, account for more than half of fatal and injurious road crashes globally [1–5]. However, the prevalence of VRU crashes is region-dependent. In Japan, VRU crashes comprise 51% (pedestrians 36% and cyclists 15%) of all fatal crashes, which is high compared to the UK with 25%, France with 16%, and Germany with 15% [6,7]. In the USA, 6827 VRU fatalities (5987 pedestrians and 840 cyclists) were reported in 2016, representing 16% of all road fatal crashes that year and the highest rate since 1991 [8,9].

The likelihood of motor vehicle-to-VRU collisions is affected by the amount and speed of motor vehicles, roadway design, and mobility habits [10,11]. The proportion of crashes causing severe and fatal injuries to VRUs in urban environments is higher than in inter-urban roads and highways [12]. However, the severity of crashes in inter-urban roads and highways with no segregated infrastructure for VRUs is higher than in urban areas due to the higher motor vehicle speeds [13]. In developed countries, the number of cyclists fatally or severely injured in motor vehicle crashes has increased with increased cycling habits [14–16]. A similar trend is observed in The Netherlands [17,18], New Zealand [19], and Sweden [20,21]. In the USA, while the total number of traffic fatalities has steadily decreased between 1994 and 2018, the proportion of pedestrian deaths has increased from

13.4% to 17.2% within the same period [22]. This proportion is projected to continue increasing [23]. Therefore, additional efforts to continue improving road safety for all road users are required in general and for VRUs in particular [24].

In the highways, the lack of segregated spaces such as sidewalks or safe walking segments forces pedestrians to walk on the side or shoulders of the roadway with no safe clearance distance to the passing traffic [25]. Pedestrians are always advised to be attentive and face the traffic [26], but even when they follow the recommendations, their safety is highly dependent on drivers' attentiveness and performance. Given the higher vehicle travelling speed on highways and the longer time and distance required to stop the vehicle, compared to urban roads, the fatality rates of VRU crashes on highways are at least twice that of urban roadways [27]. Bad weather, poor lighting conditions, driver distractions [28], and delayed responses due to long- and aimless highway driving [29] are factors that negatively affect drivers' ability to recognize VRUs on time to prevent a collision.

Driver assistance systems are already contributing to reducing road traffic crashes caused by drivers' failure to perceive and respond to road hazards and unintended lane departures [30]. Active safety systems, such as autonomous emergency braking (AEB) and autonomous emergency steering (AES), have been developed and tested at relatively limited speeds of up to 60 km/h to avoid single-vehicle, vehicle-to-vehicle, and vehicle-to-VRU collisions [31–35]. Simulation, naturalistic driving, and accident statistics studies have shown that integrating such active safety systems can reduce single and multi-vehicle crashes and is promising for vehicle-to-VRU collisions [9,36,37].

As driver assistance systems evolve into automated driving systems (ADS), the driver will eventually delegate the driving task entirely to the system, assuming the role of a passenger with no system and environment monitoring requirements [38]. Because the presence of VRUs on limited-access highways is unusual and frequently illegal, safety research on the prospective AV-VRU interaction has predominantly focused on urban environments. However, although less frequently than single-vehicle and vehicle-to-vehicle crashes and vehicle-to-VRU collisions in urban environments, accident data indicate that a considerable amount of vehicle-to-VRU collisions on limited-access highways occur every year worldwide [22,39,40]. Not addressing these cases may limit the effectiveness of road traffic-safety measures endangering VRUs and negatively affecting the social acceptance of AVs. Thus far, there is a gap in the literature concerning research on how AVs will avoid VRUs on limited-access highways [41].

ADS users are not required to monitor the roadway during automated driving and may be engaged in non-driving-related tasks when an AV encounters a VRU on its pathway. Therefore, VRU detection and avoidance are the responsibility of the system. However, AVs equipped with active safety systems (i.e., AEB and AES) have been evaluated and proven effective at speeds of up to 60 km/h. Beyond this speed, although the technology may detect a VRU [42], system capabilities of effectively avoiding an imminent collision are not guaranteed [43]. If the system fails to perceive or avoid a VRU on the roadway, ADS users with eyes off the road may also miss such an event, resulting in a crash. Therefore, data from several studies suggest that incorporating ADSs in highways should be done under specific operating speeds for which the system capabilities to perceive and avoid obstacles have been proven safe [44,45].

The ultimate goal of this review study is to contribute to VRU safety and the safe deployment of AVs. Specifically, this paper reviews the evidence for vehicle-to-VRUs collisions on limited-access highways to describe scenarios relevant to AVs interaction with VRUs on limited-access highways. The proposed scenarios may serve as functional scenarios that can be further parameterized to generate test cases for ADS safety assessments. The review starts with the definitions of VRU and limited-access highways adopted in this study, followed by a summary of the dimensions and proportions of vehicle-to-VRU collisions based on accident data from different regions and sources. Thereafter, typical vehicle-to-VRU collision patterns on highways considering the underlying motivations for the VRU to be on the highway are presented with schematic illustrations. Then, two ex-

amples of potential AV-VRU interactions on highways are discussed to illustrate how the proposed collisions patterns can be extracted and defined to account for AV testing scenarios. The paper ends with a chapter on concluding remarks.

## 2. Method

### 2.1. Study Approach

A narrative review was conducted as considered to be an appropriate approach for addressing the broad research question. The review covered academic publications, scientific and public road traffic crash reports, and in-depth case studies from several sources, including the Institute for Traffic Accident Research and Data Analysis (ITARDA) in Japan, the European Road Safety Observatory (ERSO), and reports from the USA like National Highway Traffic Safety Administration (NHTSA) and AAA Foundation for Traffic Safety [22,39,46–51]. Government documents, regulations, online multimedia, press releases, and web pages in English, Japanese, Swedish, Spanish, and German were also included in the review. Further, the review included reports and experts opinion by insurance companies and legal firms.

Using authenticated databases (i.e., Google Scholar, PubMed, Scopus, Embase, Cochrane Library, CINAHL, PsycINFO, and Web of Science), the search was conducted with both keywords and subject headings. Terms in the search string ("vulnerable road users", "pedestrian", "walkers", "walking", "cyclist", "bicycling", "traffic", "safety", "crashes", "traffic deaths", "fatality", "injury", "highway", "limited-access", "expressway", "collision warning", "avoidance", "detection", "active safety systems", "automated driving", "self-driving", "autonomous vehicle", "autonomous emergency braking", "autonomous emergency steering") were combined using Boolean search operators (AND, OR, and NOT) to control retrieval.

The preliminary search yielded more than 2082 hits that have been reduced to 830 after duplicates were removed and then filtered based on title and abstract to 308 hits. The final search produced 61 articles based on a full-text review, as outlined in Figure 1. The inclusion criteria are directly related to the research aim and question, including all sources (i.e., commentaries, opinions, reviews, reports, and research). The exclusion criteria contained search terms in a different context to the research aim and question, poor language that prevented understanding, and research papers with inadequate methodology. Table A1 of Appendix A summarizes the eligible articles, reports, and news.

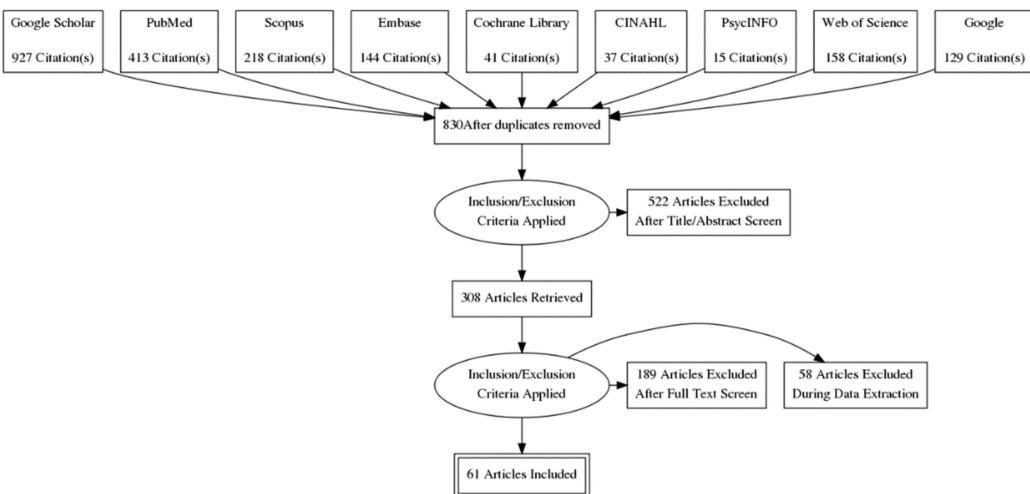

**Figure 1.** Flow diagram of the review process based on Preferred Reporting Items for Systematic reviews and Meta-Analyses (PRISMA) guidance [52]. The last search update was conducted on 1 March 2022.

## 2.2. Definition of Vulnerable Road Users (VRU)

While the potential benefits of driving automation on road traffic safety have been widely highlighted, it is essential to verify and identify the impact of these technologies on various categories of VRU prior to their deployment. The intelligent transport system directive defined VRUs as "non-motorized road users, such as pedestrians and cyclists as well as motorcyclists and persons with disabilities or reduced mobility and orientation" [2]. The definition also considers all road users that lack external protection as vulnerable.

Due to the limited evidence of motor vehicle interaction with all categories of VRU on limited-access highways, the current review predominantly focuses on pedestrians. However, the outcome of this study provides a foundation for future studies on the interaction between automated vehicles and other types of VRU on highways, such as cyclists.

## 2.3. Definition of Limited Access Highways

This study adopts a UN definition of limited-access highways (hereafter referred to as highways) as roads specifically designed to accommodate motor vehicles circulating at high speed, with lack of pedestrian infrastructures, such as sidewalk and zebra crossing, and controlled with toll gates where traffic gets on and leaves at selected locations only [26]. This definition is consistent with other international terms, such as interstate highways in the US, motorways in Europe and the UK, and expressways in Japan [53–56].

National traffic regulations and the Vienna Convention on road traffic prohibit pedestrians and cyclists from using highways [5,26]. Despite these rules and recommendations, pedestrians use highways for different reasons, exposing themselves to the risk of being hit by motor vehicles circulating at high speeds [51]. The findings of this study may also inform on safety matters relevant for other road types that lack segregated pedestrian infrastructures, such as rural highways.

## 3. Results

The proportion of VRU fatalities that occurred on highways not designed for walking is noticeable. In Japan, motor vehicle-to-VRU collisions account for 1.1% of highway crashes and 10% of highway traffic fatalities [57]. More than 800 VRUs lose their lives annually on highways in the USA, accounting for 10% of the total pedestrian fatalities [22,50,58]. In Europe, VRU fatal crashes on highways account for approximately 8% of all pedestrian fatalities [59]. Generally, the fatality rate of VRU crashes on highways is higher than on other road types as crash impact and severity are proportional to vehicle speed [60,61].

The underlying reasons for VRU to be on highways may determine their awareness and exposure to risks, their behavior, and their capacity to avoid possible conflicts with vehicles circulating at high speeds. Therefore, this chapter categorizes the review's findings considering these reasons. The review results are summarized in schematic figures that may be considered when designing scenarios to evaluate AVs' safety and interaction with pedestrians on highways.

## 3.1. Occupants Alighting from a Disabled Vehicle

Vehicles stopped on a highway due to a crash, failure (e.g., mechanical, electrical, tire burst), or running out of fuel comprise the most common reasons for vehicle occupants to alight from their vehicles on highways [22,39,48]. Alighted occupants are likely hit by other cars while trying to fix the failure on their car or to push it to the roadway shoulder. Other occupants injured and suffered from a previous crash and were thrown out of their crashed vehicles are hit by other vehicles circulating at high speeds.

In Japan, about 0.1 million vehicles annually receive road assistance on highways due to failure or crash [39], which accounts for nearly 60% of all assisted vehicles in all road types [62,63]. In comparison, roughly 9.4% of all vehicles that stop on highways require assistance after their involvement in a crash. In the USA, 18% of all VRU fatalities on highways involved occupants alighted from their disabled cars due to mechanical issues or single-vehicle crashes before they got struck by other vehicles [22].

The disabled or crashed vehicle may block the lane entirely or partially, as shown in Figure 2. When an AV encounters a stopped vehicle or a crash site on a highway, the presence of VRUs is probable. Therefore, anticipatory actions by the AV based on the sole presence of a vehicle may be considered. One such measure is to alert the AV's driver to engage and be ready for an urgent take over if the system cannot handle the situation or a time-critical condition occurs due to unexpected and sudden human movement around the incident scene. In such dangerous cases, the AV's driver may need to interrupt the automated process based on the observation and judgment of the situation. For an effective driver intervention to avoid a dangerous situation, the driver needs to be attentive (e.g., eyes on the road) and engaged in the automated process (e.g., supervising the ADS function) in advance.

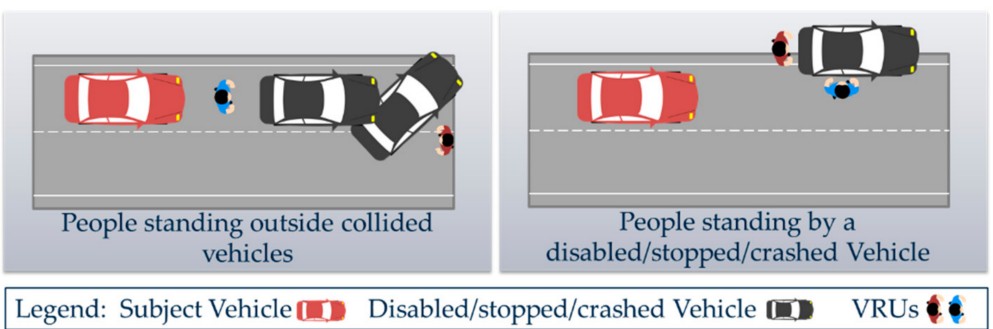

**Figure 2.** Left: Occupants alighted from or thrown out of vehicles involved in a multi-vehicle crash. Right: Occupants alighted from a disabled vehicle or a vehicle involved in a single-vehicle crash where they can be standing near or moving around the vehicle or injured and lying down on the road.

*3.2. People Walking across or along a Highway*

Cases in which people on highway lanes or shoulders are struck by motor vehicles at high speeds are also commonly reported, as shown in Figure 3 [50]. In the USA, people fatally injured while crossing the highway represent the most common vehicle-to-VRU collision pattern [50,64,65]. Concerning the location of VRUs when fatally struck by a vehicle, 77% of the crashes occurred when a VRU was on the roadway lane, 13% of the crashes occurred when a VRU was on the roadway shoulders, and the remaining 10% occurred outside the main roadway, such as ramps or the surroundings of toll gates [50].

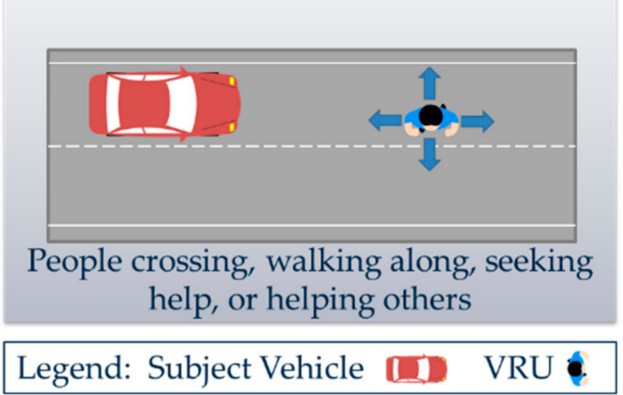

**Figure 3.** A single and non-obstructed pedestrian is standing on, crossing, or walking along the highway. People can also be seeking help, helping others, or trying to collect a fallen item on the highway.

A VRU can be moving on the highway lane or shoulders in the same direction or facing the traffic. The AV may detect VRUs based on human features and characteristics and prepare for collision avoidance maneuvers. Vehicle-to-pedestrian technologies, such as

vehicle-to-smart phone communication, may also help secure real-time communication to detect and understand VRU activities, facilitating AV-to-VRU interaction [66].

### 3.3. People Seeking Help or Helping Others

VRU may be on a highway lane or shoulder seeking help from other road users or helping other people in trouble [22,57,62], as shown in Figure 3. In some cases, occupants alight from their vehicle onto the highway shoulder and try to safely walk to the nearest exit or contact point to get help. In others, drivers stop on the opposite side and attempt to cross the highway in a risky manner to reach the site of interest [48]. Car occupants may also stop their vehicle and step out onto the highway to inspect crash damage or rescue other injured people [67].

Safety concerns of AV-to-VRU interactions presented in the first and second categories above apply to this category. However, because people may carelessly expose themselves to traffic on a highway lane to reach the target quickly, these scenarios may be more time-critical and challenging in terms of perception and reaction time demands for the ADSs.

### 3.4. People in Working Zones

Encountering a blocked highway lane due to road maintenance or construction frequently occurs [22,57,62,68], as shown in Figure 4. Drivers should be cautious and reduce speed when passing a working zone. Work zone risk in increasing crash probability in highways has been established and investigated [69]. Bumping into equipment in a road maintenance zone may endanger workers inside that zone. Although roadway workers are trained not to expose themselves to passing traffic, they may occasionally use the opened lane to avoid moving through the road maintenance equipment, endangering themselves to be struck by moving vehicles.

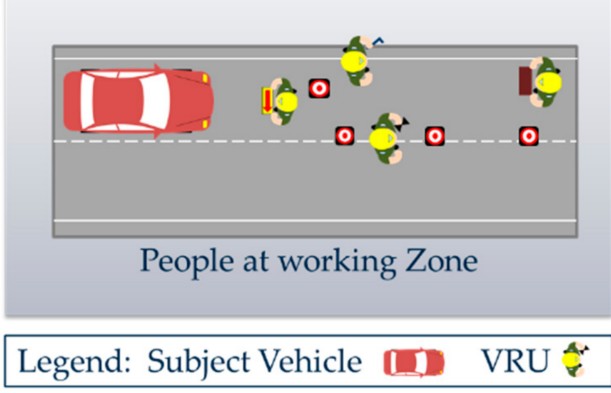

**Figure 4.** Roadway workers are standing by or moving around structures in a work zone.

There is concern regarding AV's ability to detect road works, particularly blocked lanes [70]. An entire lane can be blocked, and vehicles are obliged to change lanes. An AV may encounter different signs on its driving lanes, such as an arrow, cone, or a person with a flag, urging vehicles to change lanes. Although such scenarios may not be time-critical, AV's failure to detect working zone structure and nearby people may result in a vehicle-to-VRU collision.

### 3.5. Occupants Moving Outside Stopped Vehicles

In some cases, drivers decide to stop and alight from their vehicle on highway lanes. Contributing reasons may include picking up an item that fell off the vehicle, impairment due to alcohol, drugs, and suicidal behavior [57,65,71]. Particularly, when a person walks along the highway after dark or during bad weather conditions, for any reason, the likelihood of being hit by other inattentive drivers increases dramatically. VRU behavior and driver reaction in such cases may be exceptionally unpredictable and dangerous [50].

In other cases, police officers and medical staff may also be on the highway performing their duties (Figure 5). No official reports containing overall statistics of police and medical staff casualties during traffic stops or accidents on highways were found in this review, but such cases are commonly reported [72–75]. The AV's ability to handle such scenarios is crucial to ensure automated driving safety on highways as well as to contribute to social acceptance [76]. Compelling AV-to-others connection strategies, including the police, may be considered to avoid potential conflicts between AV and VRUs [77].

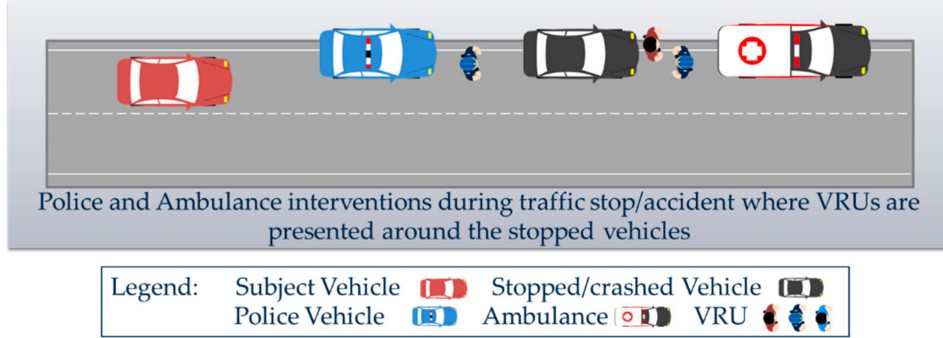

**Figure 5.** Occupants alighted from a police car or ambulance may stand near or move around the vehicles during a police traffic stop or crash involving injured people.

## 4. Discussion

The results of the current paper can inform ADS developers, standardization bodies, and policymakers toward the development of improved AV-to-VRU safety assessment strategies that complement the currently addressed by international regulations and standards. For example, the UN R157 [78], the first international regulation ever on a level 3 Automated Lane Keep Systems for highways, reads: "The activated system shall avoid a collision with an unobstructed crossing pedestrian in front of the vehicle". However, no requirement is provided concerning pedestrians partially obstructed by disabled or crashed vehicles, pedestrians interacting with other pedestrians in seek of help, or maintenance, police, or medical staff conducting their duties on highways, all of which were found very relevant in the current review. Similarly, the current standard ISO DIS 34502 [79], which focuses on establishing a scenario-based safety evaluation framework for level-3 (partial driving automation) and above ADS, incorporates pedestrians as perception targets and the need to address the perceived targets in a generic over-simplified manner safely. However, none of them accounts for the complexity of the VRU characteristics in their interaction with other participants or objects commonly present in the accident scene.

Although highways are not designed for walking, pedestrian crashes occur every year [5,7,8]. The presence of VRUs on highways occasionally occurs due to different reasons, the most common being occupants alighted from disabled vehicles [22,57], suggesting that a stopped vehicle on a highway may be an indicator to consider anticipatory preventative measures (e.g., speed reduction) based on the likelihood of sudden VRU appearance. Nevertheless, the presence of VRUs that is not related to stopped or crashed vehicles is another significant pattern, and thus unobstructed VRU also needs to be accounted for by the system.

Drivers may not expect to see people on the highway due to monotonous and high-speed driving; therefore, focusing their attention on surrounding vehicles and endangering VRUs. External factors, such as poor light during the night and bad weather conditions, can also affect driver performance and response to hazardous situations. Highway drivers usually encounter VRUs at high speeds, over 60 km/h, making human detection, judgment, and control actions challenging due to time criticality. Consequently, the fatality rate of VRU crashes on the highway is two to six times higher than in urban roadways [39], with mortality rates of 90% for average impact speeds of 80 km/h [50]. Collision avoidance systems, which have been developed to support drivers in avoiding VRU collisions, can

significantly improve traffic safety. However, thus far, the effectiveness of these systems has been predominantly proved at low speeds (<60 km/h), and their effectiveness at the high speeds typical of highways needs to be carefully evaluated.

Active safety systems (e.g., AEB and AES) serve as key enablers for ADS objects and events detection and response subtasks of the dynamic driving task. They will affect how automated vehicles drive and respond to safety-critical situations. At high-speed ranges, the performance of active safety systems may deteriorate, affecting the AEB and AES capabilities of effectively avoiding imminent collisions. For example, in a reported crash on a highway in Taiwan, the Tesla was driving on auto-pilot at a speed higher than 100 km/h when it encountered a flipped-over truck blocking the highway lane due to its involvement in a single-vehicle crash [80]. The truck driver was standing in the middle of the highway lane, moving his arms, and trying to direct the fast-approaching vehicles away from hitting the truck, but the Tesla system failed to detect him. The Tesla driver also failed to respond promptly and bumped directly into the truck. The truck driver could move out of the collision area before the impact.

This study was limited by the absence of real traffic databases and the lack of information on vehicle-to-pedestrian collisions on limited-access highways. Such real-world driving data can be collected with infrastructure sensors, drones, and test vehicles to generate scenarios for the ADS safety assessment. However, to discuss potential interactions between AVs and VRUs, the results of motor vehicle-to-VRU crashes on highways are exemplified with two traffic situations considering the AEB and AES capabilities to avoid crashes with VRUs.

Example-1: Steering away from Pedestrian

Motor vehicles overtaking pedestrians account for 10% of all pedestrian fatalities in Europe [47,81,82], and 24%, 27%, and 26% of all types of fatal pedestrian crashes in the UK, USA, and China [83–85]. AES systems are designed to steer the vehicle automatically around the detected obstacle in front to avoid a likely collision. A four-phase model (approaching, steering away, passing, and returning) typically applied to describe overtaking cyclists [86] may be adapted to model an AV overtaking VRUs on a roadway including highways, as shown in Figure 6 (top). The approaching phase starts with an AV detecting a pedestrian on the roadway. During the approaching phase, the system should perceive VRUs and decide when to steer the vehicle away based on safety indicators, such as time-to-collision (TTC), and parameters correlation, such as velocities, distance, and direction of travel ($V_{v-x}$, $V_{v-y}$, $V_{p-x}$, $V_{p-y}$). Traffic on the adjacent lane should also be considered to decide whether the steering maneuver can immediately be performed under the current speed (i.e., flying overtaking maneuver) or the vehicle speed should be adjusted (e.g., reduced) to avoid conflicts with surrounding vehicles. The AV shall move straight forward in the passing phase to overtake a VRU with a safe minimum lateral distance (MLD) between them. The returning phase starts when the ADS steers the vehicle in the direction of the original lane. The overtaking maneuver ends when AV returns entirely to its original lane and resumes driving straight forward.

Although pedestrian overtaking scenarios are not time-critical if the AV can detect the existence of the pedestrian with a TTC value equal to or more than 3 s [87,88], more vehicle-to-pedestrian collisions occurred while pedestrians were walking along with the same or opposite direction of traffic than pedestrians crossing in front of a vehicle path. According to Euro-NCAP, the latest time the system should warn the driver is 1.7 s [89]. However, naturalistic and field studies indicated that while average drivers were able to overtake pedestrians with TTS of less than 1.7 s, the TTC at which the vehicle starts steering away is highly dependent on vehicle velocity [90].

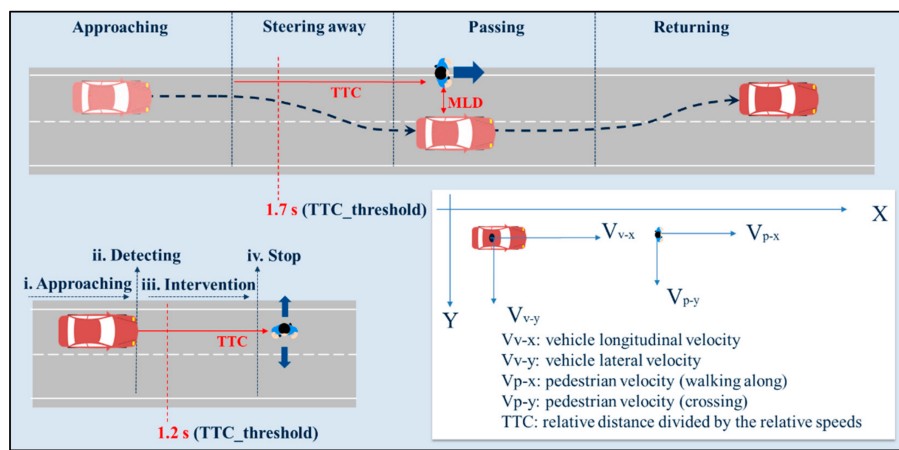

**Figure 6.** Example scenarios of potential AV-VRU interactions on highways and the effectiveness of active safety systems. (**Top**) VRU overtaking for safety evaluation of Automatic Emergency Steering systems. (**Bottom**) VRU collision avoidance for safety evaluation of Automatic Emergency Braking systems. TTC represents the remaining time before AV strikes VRU if they continue to travel with the same speed and direction.

Example-2: Braking for Pedestrian

Pedestrians crossing a highway expose themselves to great danger. They may presume that highway drivers can recognize them just like on other general roadways. In contrast, highway drivers may not expect pedestrians to endanger themselves and cross the highway. However, even with the driver detecting a pedestrian crossing the highway, some collisions may not be avoided due to the high vehicle speed. Approximately 50–75% of vehicle-to-VRU collisions are considered foreseeable, meaning that the pedestrian could be detected and the drivers braked before the impact [89].

Pedestrians crossing highways may tend to move faster than in urban environments (i.e., 3–5 km/h) to accommodate their movements to the high-speed traffic. Such fast movement in or out of the collision area may significantly conflict with the high-speed vehicles creating a highly time-critical and dangerous situation, which requires a prompt reaction [90]. For this, AEB intervention designed to brake autonomously for pedestrians crossing the vehicle's path should be quick [91].

The ability of an ADS to avoid pedestrians during automated driving depends on the system's effectiveness in performing the objects and events detection and response subtask. Current AEB technologies are estimated to reduce pedestrian fatalities by 15–30% [92]. These AEB technologies have been proven effective at operational speeds ranging between 10 and 60 km/h. Therefore, the effectiveness of AEB interventions to avoid collisions on highways can be lower than urban roadways due to high traffic speed.

As shown in Figure 6 (bottom), the criticality of such scenarios is proportional to vehicle speed and the detection time while approaching a VRU on highways. The TTC represents the time between VRU detection and the vehicle-to-VRU contact point. According to Euro-NCAP, the TTC threshold at which the system should detect the pedestrian and warn the driver is 1.2 s [34]. However, based on Euro-NCAP test data, a pedestrian collision could be avoided with TTC values between 0.3 s and 1 s [36,93]. Such scenarios are usually time-critical, and AV drivers out-of-the-loop may not be able to take over the control of the vehicle and avoid the collision manually if the system fails [94].

## 5. Conclusions

This study is the first to specifically investigate AV-to-VRU interactions on limited-access highways to provide evidence to support the development of automated vehicle safety assessments. On the one hand, the risk of fatal traffic crashes is the highest for VRUs on limited-access highways characterized by a lack of sidewalks and high traffic speed. On the other hand, highway automated driving systems users are not required to monitor the

roadway; therefore, collision avoidance is mainly the system's responsibility. Active safety technologies, such as AEB and AES, are essential parts of automated vehicles and are key enablers for automatic collision avoidance maneuvers.

The two provided examples of AV-VRU interactions aim at improving AEB and AES capabilities to avoid conflicts with VRU. However, there are situations where it is difficult for the system to expect and perceive VRU behavior. At higher speed ranges (>60 km/h), active safety systems may only mitigate collisions, which, for a VRU, may still result in death or severe injury. Given the limitations of active safety systems, we suggest considering AV's operational speed range according to ADS performance of objects and events detection and response subtasks in its operational domain.

These findings certainly add to our understanding of the problem's dimensions and the importance of protecting VRUs on limited-access highways. Future experimental and field observational studies may improve the understanding of ADS capabilities as a response to VRUs and drivers' behavior on highways. Thus far, AEB and AES have been evaluated separately, while both can be available in automated vehicles. Therefore, it would be of interest to investigate the overall effectiveness of AEB and AES systems during higher speed ranges on highways considering test scenarios derived from real-traffic data.

**Funding:** This work is supported by the Ministry of Economy, Trade, and Industry of Japan through the SAKURA project (https://www.sakura-prj.go.jp).

**Conflicts of Interest:** There are no conflict of interest to declare.

## Appendix A

**Table A1.** Characteristics of eligible studies by region, type, and main goals and outcomes.

| Authors and Year | Region | Targeted Population | Main Goals/Outcomes |
|---|---|---|---|
| [1] WHO, 2017 | Worldwide | Pedestrians | The report provides information for use in developing and implementing comprehensive measures to improve pedestrian safety. The outcomes contribute to implementing effective interventions that improve pedestrian safety, utilizing a comprehensive approach that focuses on engineering, enforcement, and education measures. |
| [2] The European Union (2010/40/EU) consolidated version: 9 January 2018 | Europe | Overall | The aim is to develop a framework for the deployment of Intelligent Transport Systems in the field of road transport and for interfaces with other modes of transport. It also aims to enable various users to be better informed and make safer and more coordinated use of transport networks. The outcomes contribute to provide specifications and standards that ensure a coordinated and effective deployment of Intelligent Transport Systems within the European Union. |
| [3] Boda (2017) | Europe | Pedestrians and cyclists | The study aims to develop new knowledge about driver behavior with VRU and integrate it into assessment programs, such as Euro-NCAP, to improve their scenario-based evaluation of the systems. They included the developed knowledge in a counterfactual analysis framework for safety-benefit evaluation. It is established that, during driver-VRU interaction, the moment a VRU becomes visible to the driver had the most significant influence on the driver's braking behavior. |
| [4] Sun et al. (2003) | USA | Pedestrians | The study aims to develop realistic models for driver-pedestrian interaction at an uncontrolled two-lane mid-block crosswalk. Different methodologies for modeling pedestrian gap acceptance and the motorist yield are proposed and examined in a field study. |

**Table A1.** *Cont.*

| Authors and Year | Region | Targeted Population | Main Goals/Outcomes |
|---|---|---|---|
| [5] WHO (2018) | Worldwide | Overall | The number of road traffic deaths continues to climb, and the SDG goal to halve road traffic deaths by 2020 has not been achieved. Reviewing the critical risk factors does show; however, that progress is being made in improving key road safety laws, making infrastructure safer, adopting vehicle standards, and improving access to post-crash care. |
| [6] Maki et al. (2003) | Japan | Bicyclists and pedestrians | Vehicle to bicyclist and pedestrian collisions were investigated based on national and in-depth accident data analyses and mathematical simulations in Japan. Component test procedures have been proposed for evaluating bicyclist and pedestrian safety based on the impact area and angle. |
| [7] OECD, Fertility rates (indicator) 2015 | OECD countries | Overall | The report provides statistical analyses of the total fertility rates |
| [8] National Center for Statistics and Analysis, NHTSA (2018) | USA | Pedestrians | Traffic Safety Facts obtained from the Fatality Analysis Reporting System (FARS) |
| [9] Bella and Silvestri (2021) | Italy | Pedestrians | The aim is to contribute to the development of pedestrian warning systems. |
| [10] Litman (2003) | USA | Pedestrians | The aim is to promote the benefits of walk and walkability. |
| [11] Wegman et al. (2012) | The Netherlands | Bicyclists | The study discusses the road safety problems of cycling and cyclists. |
| [12] Jacobsen (2015) | USA | Bicyclists and pedestrians | The study aims to examine the relationship between the numbers of people walking or bicycling and the frequency of vehicles to pedestrians and bicyclists collisions. |
| [14] Haworth (2019) | 17 developed countries | Bicyclists | A survey Study. |
| [16] Wisch et al. (2017) | Europe | Bicyclists | The study introduced the Use Cases derived from the car-to-cyclists crash data analysis. |
| [18] Schepers (2017) | The Netherlands | Bicyclists | The study explores factors contributing to the 80% reduction in the number of cyclists killed (predominantly bicycle–motor vehicle crashes) per billion bicycle kilometers in the Netherlands over thirty years. |
| [19] Balanovic (2016) | New Zealand | Bicyclists | A multi-phase investigation to improve cycling safety by changing motorist overtaking behavior. |
| [20] Ekström and Linder (2017) | Sweden | Bicyclists | The study aims to identify patterns among fatally injured cyclists in Sweden to suggest general improvements in cycling safety and specific traffic conditions. |
| [21] Amin et al. (2019) | Sweden | Overall | This report describes and analyzes road safety trends in Sweden. |
| [22] Retting (2017) | USA | Pedestrians | The study reports pedestrian fatalities by state and roadway type. |
| [25] Laird et al. (2013) | Ireland | Bicyclists and pedestrians | The study presents evidence on the value of pedestrian and cyclist infrastructure in rural roadways. |
| [31] Hayashi et al. (2013) | Japan | Pedestrians | The study evaluates the effectiveness of a pre-crash safety system with pedestrian collision avoidance to reduce vehicle-to-pedestrian crashes. |

**Table A1.** *Cont.*

| Authors and Year | Region | Targeted Population | Main Goals/Outcomes |
|---|---|---|---|
| [32] Lindman et al. (2010) | Europe | Pedestrians | The study presents a sophisticated method for estimating the potential effectiveness of a technology designed to support the car driver in mitigating or avoiding crashes with pedestrians. |
| [34] Euro NCAP (2021) | Europe | VRU | The report provides Test Protocols (car-to-pedestrian, car-to-bicyclist, and car-to-motorcyclist) for AEB VRU Systems. |
| [36] Schram (2015) | Europe | VRU | The aim is to develop test procedures for assessing AEB Pedestrian systems. |
| [37] Sander (2018) | USA and Germany | Overall | Real-accidents and driving data from the USA were used to compare the capacity of onboard sensing and V2X communication to save lives. Real-accidents data from Germany were utilized to simulate accidents with and without Intersection AEB using different parameter settings of technical aspects and driver comfort boundaries. Machine learning techniques were used to identify opportunities for data clustering. Intersection AEB was found to be effective in reducing accidents and mitigating injuries up to a specific limit. |
| [39] ITARDA (2014) | Japan | Pedestrians | Statistics of pedestrian crashes on limited-access highways (expressways) |
| [41] Tabone (2021) | Non-applicable | VRU | This study reports the opinion of sixteen Human Factors researchers about their perspectives on AVs and the interaction with VRUs in the future urban environment. The interviewees believed that fully autonomous vehicles will not be introduced in the coming decades and that intermediate levels of automation, specific AV services, or shared control will be used instead. They foresaw a significant role of intelligent infrastructures and expressed a need for AV-VRU segregation. |
| [42] Dollar et al. (2011) | USA | Pedestrians | The study evaluated the performance of sixteen state-of-the-art pedestrian detectors across six data sets. Results show that system performance still has much room for improvement despite significant progress, particularly the detection at low resolutions and partially occluded pedestrians. |
| [43] Combs et al. (2019) | USA | Pedestrians | The study investigates automated vehicles' potential for reducing pedestrian fatalities. The study analyzed 5000 pedestrian fatalities recorded in 2015 (FARS) and virtually reconstructed them under a hypothetical scenario that replaces involved vehicles with automated versions equipped with state-of-the-art (as of December 2017) sensor technology. |
| [44] de Miguel et al. (2019) | Spain | Pedestrians | The study evaluated pedestrians' interaction with level-5 automated driving vehicles on public roads. |
| [45] Gelbal et al. (2020) | USA | Pedestrians | This study evaluated pedestrian collision avoidance systems for low-speed autonomous shuttles based on Vehicle-to-Pedestrian (V2P) communication. |
| [46] European Commission (2017) | Europe | Overall | Real-accidents data report |
| [49] Hu and Cicchino (2018) | USA | Pedestrians | The study investigates how pedestrian fatalities trends vary by roadway, environmental, personal, and vehicle factors. |
| [50] Wang and Cicchino (2020) | USA | Pedestrians | The study investigates the characteristics of pedestrian crashes on interstates and other freeways in the United States. |

**Table A1.** *Cont.*

| Authors and Year | Region | Targeted Population | Main Goals/Outcomes |
|---|---|---|---|
| [51] Hunter (2020) | USA | Pedestrians | This case study aimed to determine the causes of pedestrian crashes on interstate highways and potential countermeasures to reduce the crash rate for these accidents. |
| [57] Harruff (1998) | USA | Pedestrians | The study performed a retrospective analysis of 217 pedestrian fatalities in Seattle over six years using medical examiner records with essentially all deaths examined by autopsy. |
| [58] Johnson (1997) | USA | Pedestrians | The study identified crash types and factors contributing to fatal pedestrian crashes on Interstate highways and surveyed countermeasures that address the problem. |
| [59] Cieslik et al. (2019) | Europe | VRU | The project (PROSPECT) aims to improve the effectiveness of VRU avoidance systems compared to those currently on the market by expanding the scope of urban scenarios addressed and improving the autonomous emergency braking and steering systems. |
| [60] Rosén and Sander (2009) | Sweden | Pedestrians | The study developed an improved risk function for adult pedestrians hit by the front of passenger cars based on the most extensive in-depth pedestrian accident study undertaken to date. |
| [61] IIHS-HLDI (2021) | USA | Pedestrians | Fatality facts report |
| [62] Japan Automobile Federation (2020) | Japan | Motorized Vehicles | An annual report of vehicles required service on roadways. |
| [67] Officer Magazine (2020) | USA | VRU | News: A police officer was hit in a highway crash. |
| [68] Andersson and Chapman (2011) | UK | Overall | This study investigated the impact of weather factors on road maintenance and traffic accidents rate. |
| [69] Li and Bai (2009) | USA | Overall | The study reports the impact of the work zone risk factors on the probability of fatalities when severe crashes occur based on a screening process that incorporates both statistical analyses and empirical research findings. |
| [71] Centers for Disease Control and Prevention (1994) | USA | Pedestrians | The report uses FARS data to characterize intoxicated pedestrians older than 14 years killed in motor-vehicle-related crashes. |
| [72] kiiitv.com (2020) | USA | VRU | News: A police officer was hit in a highway crash. |
| [73] nbcboston.com (2020) | USA | VRU | News: A police officer was hit in a highway crash (Tesla autopilot). |
| [74] nbcconnecticut.com (2019) | USA | VRU | News: A police officer was hit in a highway crash. |
| [75] Police Magazine (2020) | Australian | VRU | News: A police officer was hit in a highway crash. |
| [79] ISO WD34501 (2021) | Not applicable | Overall | Automated Vehicle Standardization |
| [80] taiwannews.com (2020) | Taiwan | VRU | News: Video shows Tesla on autopilot slam into a truck on Taiwan highway. |

<center>**Table A1.** *Cont.*</center>

| Authors and Year | Region | Targeted Population | Main Goals/Outcomes |
|---|---|---|---|
| [81] Lübbe (2015) | Sweden | Pedestrians | This study developed an integrated pedestrian safety assessment method using data from passive safety and active systems evaluations and demonstrated its use in assessing combinations of passive and active safety systems of autonomous emergency braking and forward-collision warning.The study outcomes show that the autonomous emergency braking system has a safety benefit broadly equivalent to increasing the Euro NCAP passive safety rating. |
| [83] Wisch et al. (2013) | Europe | VRU | A project aims to improve VRUs safety by developing test and assessment procedures for forward-looking integrated pedestrian safety systems that incorporate passive safety and autonomous emergency braking systems. |
| [84] Yanagisawa et al. (2017) | USA | Pedestrians | The study estimates the effectiveness and potential safety benefits of pedestrian crash avoidance and mitigation systems in light vehicles. |
| [85] Chen et al. (2015) | China | Pedestrians | The study conducted in-depth accident analysis to describe accident scenarios for pedestrian accidents in China and to support the development of test procedures for assessing autonomous emergency braking systems. |
| [86] Kovaceva et al. (2019) | Europe | Cyclists | The study quantified drivers' comfort zone boundaries and investigated influencing factors while drivers overtake cyclists in a naturalistic setting. |
| [87] Rasch et al. (2020) | France and Sweden | Pedestrians | The study aimed to address pedestrian-overtaking maneuvers on rural roads by analyzing how drivers adjust their behavior using safety metrics extracted from field and driving simulator studies. The study analyzed and modeled the driver's comfort zone when overtaking a pedestrian. |
| [90] Brännström et al. (2014) | Sweden | Overall | The study reports an evaluation of autonomous emergency braking and steering systems. |
| [91] Euro-NCAP (2019) | Europe | Overall | The report explains the safety assessments of the autonomous emergency braking system. |
| [92] European Road Safety Observatory (2016) | Europe | Pedestrians | A traffic safety fact report. |

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
