# Peer review of "A Review of Vehicle-to-Vulnerable Road User Collisions on Limited-Access Highways to Support the Development of Automated Vehicle Safety Assessments"

_safety, 2021_

Round 1

Reviewer 1 Report

The paper presents VRUs collisions on limited-access highways for the sake of the development of automated vehicle safety assessments. This is a well-written paper containing some results which merit publication. The comprehensiveness of the data sources in the article deserves recognition. For the benefit of the reader, however, a number of points need clarifying and certain statements require further justification. There are given below.

  1. The article mentioned that VRUs is defined as “non-motorized road users, such as pedestrians and cyclists as well as motorcyclists and persons with disabilities or reduced mobility and orientation” by the intelligent transport system, but the paper predominantly focuses on pedestrians and pays less attention to other groups.
  2. Are there any similarities and differences between collisions that occur at night and collisions that occur during the day? In addition, perhaps weather factors can also be taken into consideration in the future. For instance, in rainy and snowy weather, how long braking distance and low visibility will affect the collisions?
  3. On Section 4: The serial number of picture 3 is incorrectly marked as 2.
  4. AEB and AES have been mentioned several times in Discussion and Conclusions, but not mentioned in Abstract. Meanwhile, the description of auto-mated vehicle safety assessments is somewhat sparse.

Reviewer 2 Report

The review is well written. My comments are as follows.

1. The reference investigation is not sufficient. For example, the rollover is a classical vehile danger. In the background, this issue can be investigated: 10.1504/IJVD.2021.117154, 10.1504/IJMIC.2019.101956, 10.1049/iet-csr.2019.0030

2. Check the whole paper to avoid any possible grammar errors.

Reviewer 3 Report

The reviewed article is a review work. The authors made a significant contribution to the preparation of this article. In my opinion, after the language test, it can be published in a journal. 

Reviewer 4 Report

This is review study should be significantly improved.

The research scope is rather limited, only focus on VRU for limited-access highways.

The structure is confusing. In section 2, method section, different analysis approaches should be introduced rather than the data collection.

Section 3, the authors are suggested to prepare a table to compare the difference among different studies.

Discussion and conclusions, what are the main future works for this research topic?

Reviewer 5 Report

  1. The Introduction and Method sections can be shorten, since some information is commonly known, e.g. vulnerable road users, limited access highways
  2. It is stated in the manuscript that “This study was limited by the absence of real traffic databases and the lack of information on vehicle-to-pedestrian collisions on limited-access highways”. The reviewer agrees.  Without real world data or simulation data, the findings or conclusions of the study is not concrete.     Even the two examples presented in the manuscript are again a compilation of literature findings.
  3. The Results presented five possible causes of VRU fatalities. Those cause are commonly known, the reviewer has some questions on the innovativeness of the study.
  4. Also the five causes identified are also common for non-limited access highways, the study indicated it was focused on limited access highways. The focus of limited access highways was not clear.
  5. It is stated in the manuscript “investigate AV-to-VRU interactions on limited-375 access highways to provide evidence to support the development of automated vehicle safety evaluation strategies”. After reviewing the manuscript, the reviewer found the study pointed out some shortcomings/weakness of current AV systems, but didn’t provide information on improved strategies, need to explain further or elaborate.
  6. Figure 1 is not very useful, the reviewer thinks what is more important for a review study is the synthesis of the search results, not the number of documents reviewed. And given the number records, without a comparison number, it is hard to know if the number is enough or not.
  7. The figure numbers are messed up. Also the layout of figure 2 is awkward. The reviewer thinks it would be better to break it into several figures.
  8. proof reading is needed.

Round 2

Reviewer 1 Report

The authors have made good revisions to the manuscript.

Reviewer 2 Report

Thanks for the authors’ response. This paper can be accepted since it has been improved a lot for satisfying the journal’s quality.

Reviewer 3 Report

The work has been corrected in accordance with the reviewer's guidelines. 

Reviewer 4 Report

The review comments are not well addressed by the authors. The authors did not spend time to improve the quality of the manuscript.   My suggestion is rejection.

Reviewer 5 Report

The authors address all my comments. 

This manuscript is a resubmission of an earlier submission. The following is a list of the peer review reports and author responses from that submission.

Round 1

Reviewer 1 Report

Thank you for asking me to review the article: A review of vehicle-to-vulnerable road user collisions on highways to support the development of automated vehicle safety assessments. This is an interesting article, that is clear ad well written, however I have a few concerns that suggest be addressed. In general I think there needs to be some thought around the structure i.e., the background need for a review vs. the outcomes of the review. The review outcomes seem to be framed more as ‘why and how are people involved in incidences on highways’ this is not quite the same thing as the stated aim of  “to review the literature searching for evidence to support the development of safety evaluation strategies to avoid AV-to-VRU collisions on limited-access highways”’.

P2: why specifically highways? Perhaps explore this "gap in the literature" a little more? e.g., what if different about highways that means we can't extrapolate from the literature that already exists? As I read forward, I think you do this, but I needs to be clearer up front. For example – p5 lines 193-212 might be better placed in the introduction to set up the review.

P2, L70 – you use the term “study”, it should be “review”

P3: regarding the search terms. I note that you didn’t include "self drive(ing) vehicles". Particularly important in the more 'popular' outlets as it shifts towards the vernacular rather than the 'scientific'. I realise that you don’t want to go back and do the whole review again, but perhaps you could please comment?

P3 (and also above), you point out that incidences of this type are quite rare. Again, you go into this later, but I think that teasing this out a little more needs to be front-and-centre. – for example, I would suggest that section such as p4, lines 163-168 would be better positioned in the introduction. Similarly P7, lines 267-271

P4 Lines 172-173: this is an interesting point, however I would expect that the AV will have already issued a TOR based on the presence of an incident. So I suspect that it would be more nuanced than that i.,e to be aware of pedestrians given the location and type of incident

There are also a number of examples of general AV-VRU risk that are risks anywhere, not just highways. I suggest making the thread tighter between these points and highway driving. E.g., p 6, 207-212, p7, 265-266

Lines 288-294: This is exactly the information that needs to be in the introduction to set the framework for the paper and need for the review.

Lines 269-309: Again - it is a reasonable model for avoiding VRU's under any situation - why is this specific to highways?

Author Response

Dear Reviewer

Best wishes,

HUSAM

Reviewer 2 Report

This paper addresses an interesting and useful topic about autonomous driving. After going through, the following comments came up.

Firstly, please justify that how often non-motorized Vulnerable Road Users appear on highways. 
To my knowledge, pedestrians and cyclists, the so-called VRUs, are generally not allowed to use highways.
In particular, the scenarios shown in Fig. 2 are quite uncommon.
If this could not be well justified, the presented review makes little sense.

Secondly, the contributions are not clear. Please specify them by points.

Thirdly, the paper involves too many statistical sources, while the relevant publications are not sufficiently discussed.
Now, the paper stops at how to evaluate safety. Probably, this is not sufficient.
The discussed papers are not new, and many of them are more than 10 years.
There are many recent publications proposing various strategies that are closely related to the safety issue in autonomous driving.
Below are only some examples from some top-tier journals, and the author could find even more.
Gopindra S.Nair, Chandra R.Bhat, Sharing the road with autonomous vehicles: Perceived safety and regulatory preferences, Transportation Research Part C: Emerging Technologies, 2021.
Chao Huang, Peng Hang, Zhongxu Hu, Chen Lv, Collision-Probability-Aware Human-Machine Cooperative Planning for Safe Automated Driving, 2021.
Another section to discuss these strategies is a must from my view.

Finally, the suggested way of implementing an AV speed range is quite straightforward.
It would be necessary to come up with new ideas.

Author Response

Dear Reviewer

Best wishes

HUSAM

Reviewer 3 Report

Dear Editor and dear authors,

I have really enjoyed reading a comprehensive, well-flowing and systematically conducted and presented review article. I am feeling that the manuscript is of substantial merit and it informs the state of the art in a timely and meaningful way. I liked the description of the method which was detailed and the excellent use of figures and scenarios.

I have some minor points to highlight only:

  • The work needs in the conlcusions a honest reflective commentary referring to its limitations (e.g., no primary data analysis, too many hypotheticals etc.). Acknowledge the shortcomings and being critical about your own work adds value actually. A small paragraph could fix this.
  • I need to see in the discussion a bit more on policy, planning, technology implications. How your study affects these? What is your key contribution in applied terms? A few lines would be enough.
  • Since this is a review I found hard to get the results and discussion headings; the content is not a match for these titles since they are very close to each other. Find different headings.
  • Consider these papers that you missed; they are helpful: https://doi.org/10.1080/15389588.2017.1387654 & https://doi.org/10.3390/safety4020020 

As a whole very happy of the paper's standards.

Author Response

Dear Reviewer

Best wishes

HUSAM

Round 2

Reviewer 1 Report

I am happy with this version of the manuscript

Reviewer 2 Report

Some of the comments have not been addressed satisfactorily. Impractical materials are still there.

The state of the art has not been comprehensively reviewed. Also, the reference list is quite unprofessional, e.g., I could not find ref [95], [98] seems incomplete, etc.

Take-away is little.

Reviewer 3 Report

I am okay with the paper. It is good to go. Relatively surprised that one of my only two reference recommendations was not included.